# Single-domain antibody screening by *is*PLA-seq

Yueyuan Yin*, Fei Yan*, Ruimin Zhou*, Mingchen Li, Jinyi Ma, Zhe Liu ⓘ, Zhenyi Ma ⓘ

Single-domain antibody (sdAb) holds the promising strategies for diverse research and translational applications. Here, we describe a method for the adaptation of the in situ proximity ligation assay (*is*PLA) followed by sequencing (*is*PLA-seq) to facilitate screening of a high-sensitive, high-throughput sdAb library for a given protein at subcellular and single-cell resolution. Based on the sequence of complementarity-determining region 3 (CDR3), the recombinant sdAb can be produced for in vitro and in vivo utilities. This method provides a general means to identify the functional measure of sdAb and its complementary epitopes and its potential applications to investigate cellular processes.

## Introduction

Antibodies hold the preferred strategy for various applications to expand both basic and translational researches (Carter & Lazar, 2018). The single-domain antibodies (sdAbs) with complementarity-determining region (CDR) are derived from the variable domains of heavy chain-only antibodies (Helma et al, 2015; Muyldermans, 2021). Because of their high binding capacity, stability, and specificity, sdAbs interfere with protein functions and are used as research tools for in vitro and in vivo purposes (Tanaka & Rabbitts, 2010; Al-Baradie, 2020). Yeast-two hybridization and phage surface display are usually two screening procedure of protein–protein interaction including sdAb and its recognizing epitope (Tanaka & Rabbitts, 2003). The resulting sdAb retains antigen recognition capacity and can circumvent the need for the conventional animal immunization and hybridoma isolation (Ahmad et al, 2012; Ascione et al, 2019). However, the rapid and robust screening approach for high-sensitive, high-throughput and affinity and specificity of sdAb and its native complementary epitope individually in single cells remains challenging.

To overcome this, we synthesized and screened a randomized CDR3 sdAb library against autophagy receptor SQSTM1 as an experimental model. We demonstrate that the identified sdAb is selectively towards the native and/or denatured epitope of SQSTM1, depending on its corresponding epitope. Therefore, these sdAbs described here can be used to specifically manipulate cellular processes for a broad range of biochemistry and cell biology applications.

## Results

### Strategy for screening sdAbs against SQSTM1

Using a high-throughput in situ proximity ligation assay (*is*PLA)-based cDNA screening of sdAb-epitope interactions in single cells followed by high-throughput sequencing (namely *is*PLA-seq), here we outlined a sdAb screening approach for SQSTM1 (Fig 1A). Except for its role involved in the process of autophagy, SQSTM1 is also a critical signaling adaptor protein with an intricate context-dependent manner (Moscat et al, 2016). Accordingly, we first synthesized a sdAb library containing a randomized 21-aa CDR3 domain with C-terminal 3× Flag tag which was transiently co-transduced with HA-tagged SQSTM1 cDNA into HEK293T cells followed by *is*PLA. After recovery of the PLA positive cells by FACS, the sdAb DNAs from these cells were amplified using the complementary primers flanking the sdAb vector by PCR (Fig 1B–D). Then, the recovered CDR3 DNAs derived from these sdAbs were recombinantly subcloned into the same sdAb vector for the next round of screening as illustrated in Fig 1A. After three rounds of screening followed by sequencing CDR3 coding regions, total 27 different CDR3 sequences were obtained among the 260 anti-SQSTM1 sdAbs (Table 1). One of them, clone #1, was most-represented and was selected for further functionality assay for recognizing distinct epitope of SQSTM1. *Pseudomonas* exotoxin A (ETA)-linked protein has been used as a delivery means to cell cytosol (Verdurmen et al, 2015; Mazor & Pastan, 2020). Next, we reconstructed the expression cassette sequentially containing GST, a tobacco etch virus (TEV) protease cleavage site, anti-SQSTM1 sdAb, the translocation domain of *Pseudomonas* exotoxin A (ETA), and 3× Flag epitope tag (Fig 1E, upper panel). After GST affinity purification and TEV protease cleavage, the recombinant C-terminal ETA-fused and Flag-tagged anti-SQSTM1 sdAb or its control without CDR3 (sdAb Con) in *Escherichia coli* BL21 cells was finally shown by Coomassie blue staining, with BSA as a loading control (Fig 1E, lower panel).

---

Tianjin Key Laboratory of Medical Epigenetics, Key Laboratory of Immune Microenvironment and Disease (Ministry of Education), Department of Immunology, Biochemistry and Molecular Biology, School of Basic Medical Sciences, Tianjin Medical University, Tianjin, China

Correspondence: zheliu@tmu.edu.cn; zhyma@tmu.edu.cn
*Yueyuan Yin, Fei Yan, and Ruimin Zhou contributed equally to this work

---

Figure 1. **High-sensitive, high-throughput screening for anti-SQSTM1 sdAbs by in situ proximity ligation assay (*is*PLA)-seq.**
(A) Workflow of sdAb library screening by *is*PLA-seq. A Flag-tagged sdAb library containing a randomized 21-aa CDR3 domain with C-terminal 3× Flag tag was synthesized and transiently co-transfected with HA-tagged bait into HEK293T cells. At 48 h after transfection, cells were fixed and probed by *is*PLA. The *is*PLA-positive cells were sorted by FACS followed by CDR3 DNA amplification. The recovered cDNAs can be reconstructed for another round screening until the final sequencing and functional validation. The CDR3-null vector was used as a sdAb control (Con). (B) Flow cytometric analysis of *is*PLA-positive cells compared with the sdAb control in HEK293T cells (*is*PLA control). (C) Representative positive *is*PLA signals (red dots) in HEK293T cells, compared with the control of anti-HA antibody alone. Nuclei are counterstained with DAPI (blue). Scale bar, 10 µm. (D) PCR products of the recovered *is*PLA positive HEK293T cells were evaluated by agarose gel electrophoresis. sdAb Con vector, non-transfected HEK3293T cells, *is*PLA positive HEK293T cells, and H₂O were used as templates, respectively. Right lane, DNA markers. bp, base pairs. (E) Recombinant anti-SQSTM1 sdAb

### Live-cell imaging and functionality of anti-SQTSM1 sdAb

To avoid the artificial effect in the validation of these sdAbs, we generated $Sqstm1^{-/-}$ Lewis lung carcinoma (LLC1) cells by the CRISPR-Cas9 technology as shown at DNA level as well as protein level (Fig 2A–C). These $Sqstm1^{-/-}$ LLC1 cells were used in the following assays.

For the functional assay of the recovered sdAbs against SQSTM1, we expected to validate at least one of them that could selectively recognize certain domains of SQSTM1 (Fig 3A). First, we transduced the truncated SQSTM1 along with anti-SQSTM1 sdAb DNA individually to test their specific association by isPLA. As expected, Flag-tagged anti-SQSTM1 clone #1 specifically recognized the introduced full length, C-terminal (221–440 aa), but not N-terminal region of SQSTM1 (1–220 aa), compared with the sdAb Con (Fig 3B). Second, 6× His-tagged SQSTM1 or its mutants was expressed in E. coli BL21 cells and purified by affinity chromatography on $Ni^{2+}$-chelated His-bind resin. After incubation with the purified anti-SQSTM1 sdAb clone #1, the Flag-tagged sdAb was detected in the His-tagged C-terminal region of SQSTM1 (221–440 aa) pulldown by Western blotting, but not in the His-tagged 1–220 or 1–120 aa elute (Fig 3C). Third, the physical interaction between anti-SQSTM1 sdAb clone #1 and C-terminal SQSTM1 was also demonstrated in $Sqstm1^{-/-}$ cells incubated with this sdAb followed by co-immunoprecipitation (co-IP) and immunofluorescent (IF) staining assays, showing the specificity of anti-SQSTM1 sdAb clone #1 recognizing as short as 319–440 aa (Fig 3D and E). Fourth, because SQSTM1 blockade results in impaired autophagic process, we tested whether anti-SQSTM1 sdAb clone #1 disturbs autophagic flux. Indeed, human lung adenocarcinoma A549 cells incubated with the purified anti-SQSTM1 sdAb clone #1 resulted in more LC3B-II accumulation upon trehalose (Tre, an autophagy enhancer) treatment, compared with the sdAb Con treatment (Fig 3F). The elevated SQSTM1 level upon anti-SQSTM1 sdAb clone #1 incubation indicated the inhibition of autophagic flux (Fig 3F), consistent with the observation of autophagic impairment (Moscat & Diaz-Meco, 2012; Todoric et al, 2017). Therefore, the purified anti-SQSTM1 sdAb clone #1 is selectively toward the epitope containing C-terminal region of SQSTM1 and is sufficient to impair autophagic flux via directly inhibiting intracellular SQSTM1.

Next, using anti-SQSTM1 sdAb clone #2, we also validated this sdAb recognized the denatured Sqstm1 in LLC1 cells, but not in $Sqstm1^{-/-}$ cells by Western blotting and IF staining, respectively (Fig 3G and H). The isothermal titration calorimetry (ITC) analysis is a widely used approach to directly measure the amount of heat released or absorbed during association processes of biomolecules and to quantitatively estimate the interaction affinity of two relevant binders (Lin & Wu, 2019). Last, the dissociation constant ($K_d$) value of anti-SQSTM1 sdAb clone #1 and GST-SQSTM1, 221–440 aa from ITC assay was 8.80 ± 1.39 $\mu M$, compared to the non-binding controls (Fig 3I). Collectively, this screen platform allowed us to identify the anti-SQSTM1 sdAbs in single cells and the ETA-tagged anti-SQSTM1 sdAb may serve as an immunoreagent to modulate function of intrinsic SQSTM1 and an immunodetection reagent in vitro.

## Discussion

Antibody therapy is still one of the most promising approaches to clinical management of various diseases. From years, the specificity of therapeutic antibodies remains a high unmet clinical need for improved treatment strategies despite the conventional antibody therapies have beneficial outcome (Wang et al, 2021). Therefore, the robust screening protocol for the disease-specific antibodies including intracellular antibodies has become much more demanding. The high sensitive, high-throughput sdAb library screening toolkit for in situ antibody-epitope recognitions described here provides several advantages as follows: (i) sdAbs prediction directly from the recovered CDR3 DNA sequences; (ii) visualization and individual quantification of subcellular distribution of sdAb-epitope recognitions; (iii) easy validation and reevaluation of sdAb functionality, and (iv) potential recognition both native and denature epitopes. Because the sdAbs identified here are derived from the native interaction in cells by isPLA (Söderberg et al, 2006), which is a reliable approach for validating these sdAbs in vitro and in vivo. The recombinant sdAbs share some desired properties including small size, soluble, monomeric, and stable proteins with low immunogenicity when administrated in vivo (Skerra, 2007). Because of the ETA transmembrane domain in these sdAbs, this property of the sdAbs described here also provides several advantages including large-scale production in bacterial host and high affinity and specificity by binding to intracellular target proteins. Therefore, the ETA-fused intracellular sdAb may lower side effect as an immunogen in vivo. In terms of anti-SQSTM1 sdAbs described here, the selective CDR3 recognizing native and/or denatured SQSTM1 epitopes can still be isolated individually, providing the potential in conventional biochemistry and cell biology assays.

On the other hand, because of the specific target properties, both cell surface and intracellular epitopes may be targeted by sdAbs, which mediate biotechnological applications for a broad range of diagnostic and therapeutic purposes in pathological conditions. Currently, the clinically used antibodies for cancer immunotherapy have been usually applied in antibody-dependent cell-mediated cytotoxicity (ADCC) or antibody-dependent cellular phagocytosis (Chen et al, 2020; Raybould et al, 2020; Zhao et al, 2020). According to their epitopes, the corresponding sdAbs may be bioengineered, isolated and applied over other immunoreagents. In addition, sdAb-conjugated small-molecule fluorophore or isotopes may also be performed in in vivo imaging (Muyldermans, 2021). Another advantage of sdAb is easy to redesign and modify, thus the large-scale engineered sdAb products in bacterial hosts

---

was produced in E. coli BL21 cells and was analyzed by SDS–PAGE followed by Coomassie blue staining, with BSA as a loading control. Upper panel, schematic showing the tandem fusion construct containing GST, tobacco etch virus cleavage site, anti-SQSTM1 sdAb or CDR3-null control (sdAb Con), the translocation domain of Pseudomonas exotoxin A (ETA), and 3× Flag tag. WCL, whole BL21 cell lysate with 0.2 mM IPTG induction; B, binding proteins on GSH resin; P, purified sdAb; M, protein molecular markers. Arrow, the purified sdAb clone #1 (upper) or sdAb Con (lower), respectively.
Source data are available for this figure.

**Table 1.**  **The CDR3 sequences of the enriched anti-SQSTM1 sdAb by in situ proximity ligation assay-seq.**

| # | DNA sequence (5′-3′) | Amino acid sequence | Frequency |
|---|---|---|---|
| 1 | gagggtctggtttactggaccactaaaaagtcgtttggaggttgtttgttacatgattggtca | EGLVYWTTKKSFGGCLLHDWS | 49/260 |
| 2 | cgggccgggtttgtggagtcgtaggttcattttccatgtacacaatcgggaactgggtctat | RAGVCGVVGSFSMYTIGNWVY | 32/260 |
| 3 | cgcgttcgcacttcccctcgcagtgcggtcgcagatgggagcgcataccgttcggagtctgcc | RVRTSPRSAVADGSAYRSESA | 26/260 |
| 4 | gcttacggcaatgacgtccagtccgtggccttcaagtatgttgtagggcgaggctaccagctt | AYGNDVQSVAFKYVVGRGYQL | 21/260 |
| 5 | tatgatagggttggtggcggtcacgtgctgattgcctgctacagtgatgacctatgtatggggg | YDRVGGGHVLIACYSDDLCMG | 13/260 |
| 6 | ctattggtaattgactcatctgttatcgcccggttagacgaccgtggtcgttctgatagattg | LLVIDSSVIARLDGRGRSDRL | 13/260 |
| 7 | gcctatattgcgaagcggccatgctgcgattgccgggaggtccaatggggcagccgcacgtgc | AYIAKRPCCDCREVQWGSRTC | 12/260 |
| 8 | gcgcgggtctttatgccatgtactccttcgcacattatcggtgcaatgtgtacattggcatgc | ARVFMPCTPSHIIGAMCTLAC | 12/260 |
| 9 | gatcgggggtgtcatagtacgttggggtcgcggatatccttttcaaaggctgtataatcggg | DRGCHSTLGSRISFFKGCIIG | 11/260 |
| 10 | gttttacatgtcctttcttgggatcggtgtgagtacgaccccgctcacgacttgcgcacggct | VLHVLSWDRCEYDPAHDLRTA | 10/260 |
| 11 | gcgcgtgcgcacttcgcctcgctctgcgactgtcaaggcaaccgcataacgctcttatgcgac | ARAHFASLCDCQGNRITLLCD | 9/260 |
| 12 | gatcaagttcgtgaccattcatggtgcaacctatttaggacattaagagtatgcgtgttaatt | DQVRDHSWCNLFRTLRVCVLI | 9/260 |
| 13 | cgggaaccatttgctcggtatctaagaaaagagtcataccctctttttcgaatgtacagtttc | REPFARYLRKESYPLFRMYSF | 8/260 |
| 14 | gggggcaacaccgtgaaggtgtccaacattttcgaagtatcgtcgtctgtgcgcgttcgtttg | GGNTVKVSNIFEVSSSVRVRL | 6/260 |
| 15 | cttcgatggagctggttcccaagcggatcaacagcgaatctatctctggggataggctgtacg | LRWSWFPSGSTANLSLGIGCT | 6/260 |
| 16 | aacatggccattagggtcaggtgggtggcaatgtttacgtgcggcccgatcggagggggtaggt | NMAIRVRWVAMFTCGPIGGVG | 5/260 |
| 17 | gcggtggaaagttgtgacgggtttcgtacaggtcggactgttccgtatcgagaggtaacgtac | AVESCDGFRTGRTVPYREVTY | 3/260 |
| 18 | ggcattgggtatgccggcgactgccgtgctcatacaattgctacgcgtaggttcagcgttgga | GIGYAGDCRAHTIATRRFSVG | 3/260 |
| 19 | agacaggcgccgttcggatcgagatgctcaatcagctcttcacctaagggacgttatccaagc | RQAPFGSRCSISSSPKGRYPS | 2/260 |
| 20 | ccaggtcacggccggccatggcgtgcggccacgtttccagtggatgaaggtggcggttcagcc | PGHGRPWRAATFPVDEGGGSA | 2/260 |
| 21 | gacctttgggctccgttgatggtgtgctctggcggtcttacccagatgttatcgaatagtgat | DLWAPLMVCSGGLTQMLSNSD | 2/260 |
| 22 | gcatgcgaactagatgctgttaggctttggccggactttgtgagacctgctgtatttctatc | ACELDAVRLWPDFVRPAVFSI | 1/260 |
| 23 | cggcgcggttttgtccgacacacaaaaatcaagttacttgattatgcgttaactgaagtgaga | RRGFVRHTKIKLLDYALTEVR | 1/260 |
| 24 | gtaagaacgccggtcattctcagatgttttcaatccggtattttttatttagtcggtgaagtt | VRTPVILRCFQSGIFYLVGEV | 1/260 |
| 25 | cctggtgatggtcaagaggttatatttgccttcgttcaacgctcgacccgtgcgtggcccgtc | PGDGQEVIFAFVQRSTRAWPV | 1/260 |
| 26 | tttccgtcacgtctgaagtgtacgcgcggatgtggttgcagaacgtttaactttcccgccctc | FPSRLKCTRGCGCRTFNFPAL | 1/260 |
| 27 | ataattgcagaactttgttcagggctgccgtgtacttctttcttgaggactccttgtcctgtc | IIAELCSGLPCTSFLRTPCPV | 1/260 |

can be feasible for translational purpose, lowering the invest cost of further applications. Last, the combination of the assembled sdAbs against multiple epitopes would enhance the capture of extra-cellular and intracellular targets, without loss of specificity. Based on these merits, sdAbs are expected to be favorable for a wide range of applications in biochemistry and cell biology as well.

In summary, we have developed a high-sensitive, high-throughput sdAb library screening approach to segregate native epitope with high specificity at subcellular and single-cell resolution, which may promote a broad range of applications for diverse basic research and therapeutic purposes.

# Materials and Methods

## Chemicals, enzymes, and antibodies

The following reagents were used: 4,6-diamidino-2-phenylindole (DAPI) (D9542; Sigma-Aldrich), BCA protein assay kit (23250; Thermo Fisher Scientific), BSA (New England BioLabs), D-(+)-trehalose dihydrate (T9531; Sigma-Aldrich), ECL detection reagents (32106; Thermo Fisher Scientific), EtBr (E1385; Sigma-Aldrich), GSH (G2451; Sigma-Aldrich), imidazole (I5513; Sigma-Aldrich), immunoglobin G (IgG) (AC011, mouse; Abclonal), polyethylenimine (PEI) (Polysciences), paraformaldehyde (PFA) (158127; Sigma-Aldrich), poly-lysine (P4707; Sigma-Aldrich), protein G-Sepharose CL-4B beads (17–0618-01; GE Healthcare), protease inhibitors cocktail tablet (4693132001; Roche), restriction enzymes including EcoRI and BamHI (New England BioLabs), and RIPA buffer (#9806; Cell Signaling Technology). Antibodies including anti-Flag (F3165, mouse mono-clonal; Sigma-Aldrich), anti-HA (C29F4, rabbit monoclonal; Cell Signaling Technology), anti-Flag M2 affinity beads (A2220; Sigma-Aldrich), and anti-SQSTM1/p62 (ab56416, mouse monoclonal; Abcam; #7695, rabbit polyclonal; Cell Signaling Technology, respectively) were used in this study. pEASY-Basic Seamless Cloning and Assembly Kit (CU201; TransGen Biotech), Trans2K Plus II DNA Markers (BM121; TransGen Biotech) and Page Ruler Prestained Protein Ladder (26616; Thermo Fisher Scientific) were also used in this study.

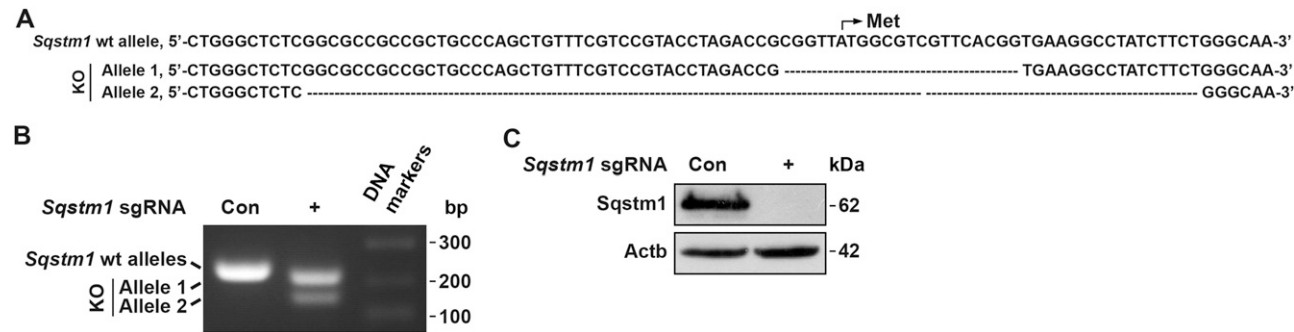

**Figure 2. Generation of *Sqstm1*⁻/⁻ LLC1 cells.**
**(A)** DNA sequences of wild type (wt) and knockout (ko) allele 1 and allele 2 of *Sqstm1* by CRISPR-Cas9. **(B)** PCR products of wt and ko alleles of *Sqstm1* in LLC1 cells were evaluated by agarose gel electrophoresis, compared with the sgRNA control (Con) cells. Right lane, DNA markers. bp, base pairs. **(C)** Validation of Sqstm1 depletion by Western blotting in *Sqstm1*⁻/⁻ LLC1 cells, compared with the sgRNA Con cells. Actb as a loading control.
Source data are available for this figure.

## *Sqstm1*⁻/⁻ LLC1 cells

*Sqstm1* knockout LLC1 cell line by the CRISPR/Cas9 technology was generated as described (Ran et al, 2013). sgRNAs were designed using sgRNA design tool (https://zlab.bio/guide-design-resources). Briefly, two sgRNA sequences, 5′-ATTAAT-GATATCTCCCGGGT-3′ and 5′-CCGTACCTAGACCGCGGTTA-3′ to target *Sqstm1* were cloned into PX458 plasmid individually (Plasmid #48138; Addgene). PX458 vector was used as an irrelevant control (sgRNA Con). Then CRISPR vectors were transduced into LLC1 cells. 48 h later, GFP positive cells were individually sorted into 96-well plates by FACS. After 2–3 wk expansion, genomic DNAs of the clonal cells were examined using PCR with the primers, 5′-CTCTTGTGGTCACCCA-TGTATT-3′ and 5′-GGCTGAAGCAGAAGCTGAA-3′ to target mouse *Sqstm1*.

## sdAb library construction

The sequence of cAbBCII10 was used as template of cloning backbone for sdAb construct (Saerens et al, 2005). The original CDR3 region (VRGYFMRLPSSHNFRY) was replaced by a spacer (5′-GAATTCGGCAGCGGATCC-3′) containing restriction enzyme cutting sites of EcoRI and BamHI. The DNA fragment of sdAb construct was cloned into pcDNA3.1 vector with a C-terminal 3× Flag tag to create CDR3-null sdAb control (sdAb Con) vector. The single-strand CDR3 DNAs containing degenerated 63 nucleotides were synthesized by Invitrogen with the sequence 5′-CTATTTAT-TATTGTGCTGCT(NNN)₂₁TGGGGTCAAGGTACTCAAGTTACT-3′, and the complementary strand was synthesized using DNA polymerase for DNA strand extension with the primer of 5′-AGTAACTTGAG-TACCTTGACC-3′. All the constructs were verified by DNA sequencing. The linearized sdAb Con vector by EcoRI and BamHI and the synthesized double-strand CDR3 DNAs were mixed for recombinantly cloning using pEASY-Basic Seamless Cloning and Assembly Kit. After transformation and amplification in *E. coli* DH5α cells, 20 clones were sequenced to confirm the random recombination. The sdAb library capacity was approximately 2.8 × 10⁵ CFU. The plasmid DNA was purified using Plasmid Midi kits (12145; QIAGEN) for subsequent screening assay.

## Cell culture and *is*PLA

HEK293T cells, LLC1 cells, and A549 cells were obtained from American Type Culture Collection and were cultured according to the recommended procedures. The Flag-tagged sdAb library was co-transfected with HA-tagged *SQSTM1* construct into HEK293T cells. 48 h after standard cultivation, the trypsinized HEK293T cells were fixed by 1% PFA and were collected for *is*PLA. *is*PLA was performed using the Duolink In Situ Red Starter Kit Mouse/Rabbit according to the manufacturer's instructions (DUO92101; Sigma-Aldrich) as previously described (Söderberg et al, 2006). Briefly, the collected cells were permeabilized in PBS containing 0.5% Triton X-100 for 10 min. Samples were incubated with blocking solution for 1 h at 37°C in a 1.5-ml tube and then 60 min at 37°C with an anti-HA rabbit monoclonal antibody and anti-Flag mouse monoclonal antibody. Cells were then incubated for 60 min at 37°C with a mix of the MINUS (anti-mouse) and PLUS (anti-rabbit) PLA probes. Hybridized probes were ligated using the Ligation-Ligase solution for 30 min at 37°C and then amplified using the Amplification-Polymerase solution for 100 min at 37°C.

## FACS analysis

After *is*PLA, HEK293T cells were washed twice with PBS buffer containing 1% BSA, 2 mM EDTA, and 0.1% NaN₃ and immediately acquired on a BD FACS Aria II flow cytometer and analyzed using FlowJo V10.0.7 software (Tree Star). The sorted PLA-positive cells were collected as PCR templates for CDR3 fragments amplification.

## PCR amplification and sequencing

The enriched cells by FACS were incubated in a 95°C block heater for 10 min and the sample was used as a template for PCR amplification with the primers of CDR3-forward: 5′-ACACCGCCATC-TACTACTGC-3′ and CDR3-reverse: 5′-GCTGCTCACTGTCACTTGTG-3′. Phusion Hot Start II High-Fidelity PCR Master Mix (Thermo Fisher Scientific) was used according to the manufacture's instruction. Up to 500 cells were used as template in 50 μl reaction mixtures containing 0.5 μM of each primer. The PCR procedure is 98°C for 60 s, 50 cycles of 98°C for 30 s, 62°C for 30 s, and 72°C for 30 s, and

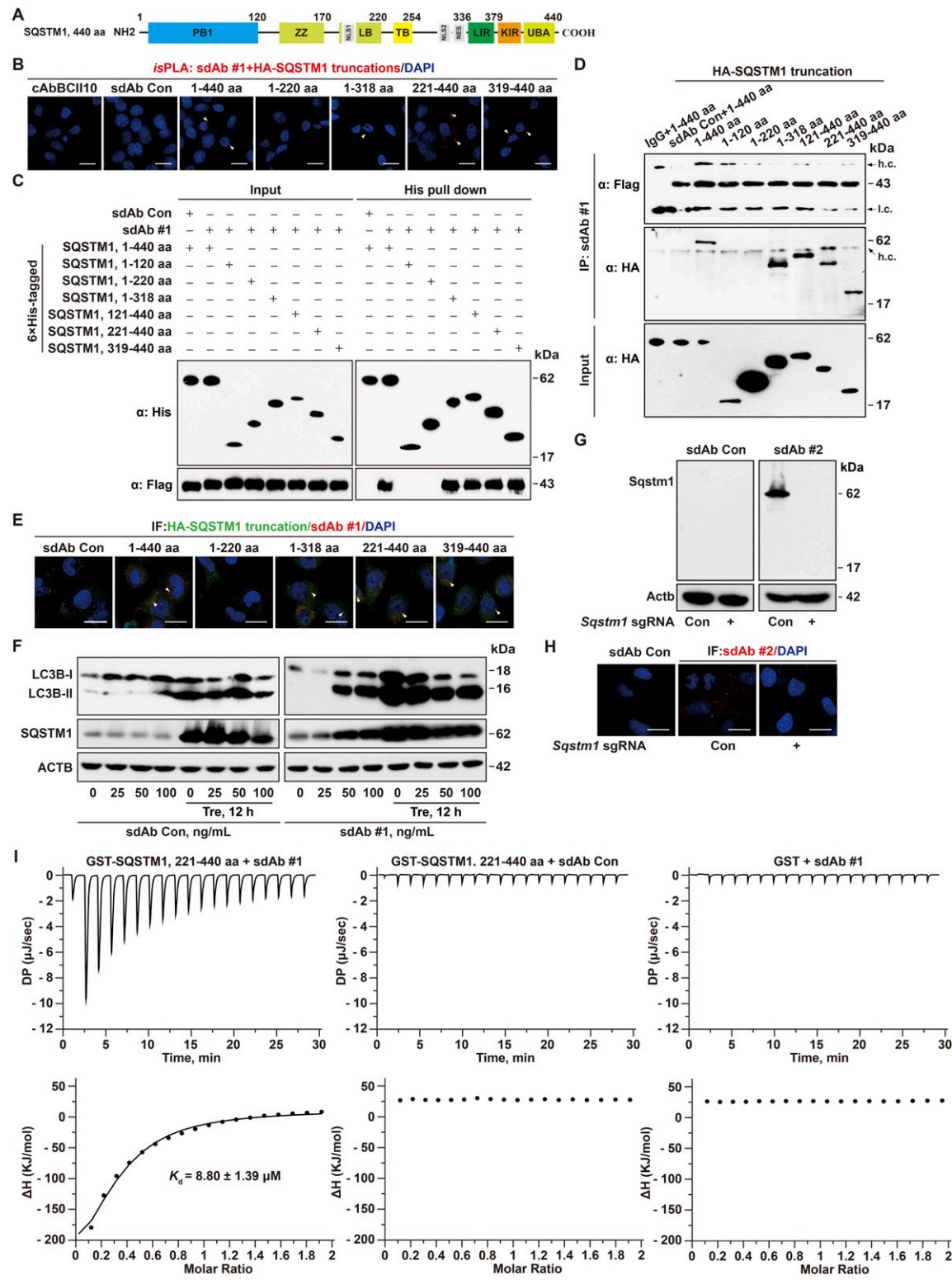

**Figure 3. Functional validation of Flag-tagged anti-SQSTM1 sdAb.**

**(A)** Schematic representation of the domains of SQSTM1 and its ΔLIR. PB1, a Phox-BEM1 domain; ZZ, ZZ-type zinc finger domain; LB, LIM protein binding domain; TB, TRAF6-binding domain; LIR, LC3-interacting region; KIR, KEAP1-interacting region; UBA, ubiquitin-associated domain; NLS, nuclear localization signal; NES, nuclear export signal. **(B, C, D, E)** Functional validation of anti-QSTM1 sdAb clone #1 by in situ proximity ligation assay using anti-HA and anti-Flag antibodies at 48 h after transfection as indicated by white arrow heads (B), recombinant protein pulldown by Ni$^{2+}$-chelated His-bind resin followed by anti-Flag Western blotting (C), co-IP detection of SQSTM1 and sdAb clone #1 as indicated (D), and co-localization by immunofluorescence staining (IF) as indicated by white arrow heads (E) compared with cAbBCII10 and sdAb Con

a final 10-min extension at 72°C. PCR products were used for the next round of cloning into sdAb vector followed by transforming DH5α or for CDR3 sequencing of individual clones. The sequencing primer is 5′-GCACCAAAATCAACGGGAC-3′. After high-throughput sequencing, docking of the recovered sdAbs and SQSTM1 molecules was conducted using HDOCK as described previously (Yan et al, 2020).

### Agarose gel electrophoresis and next round CDR3 sublibrary construction

Standard agarose gel electrophoresis was performed according to the previous description using a gel running buffer of TAE (40 mM Tris-acetate and 1 mM EDTA, pH 8.5). Briefly, EtBr was added to the gel before electrophoresis to a final concentration of 0.5 μg/ml. DNA samples were loaded into the gel wells followed by separation at 80 V for 1 h 30 min. Then, the gel was exposed to UV light and the images were taken on a gel documentation system. Last, the PCR products were purified by gel extraction kit (QIAGEN). The 122-bp CDR3 fragment amplified from the positive cells was recombinantly subcloned into the linearized sdAb Con plasmid to create sdAb sublibrary for the next round screening or the CDR3 sequences were individually determined by sequencing.

### Recombinant protein expression, purification and pulldown assay

For recombinant protein expression in *E. coli BL21* (DE3) cells, 6× His-tagged SQSTM1 or its mutants or TEV protease was individually cloned into pET28a vector (Novagen). The tandem fusion construct of sdAbs were cloned into pET32a vector (Novagen) containing GST, a TEV protease cleavage site, anti-SQSTM1 sdAb, the translocation domain of *Pseudomonas* exotoxin A (ETA), and 3× Flag tag. All the constructs were verified by DNA sequencing. The following recombinant protein purification and pulldown assay were performed as previously described (Yan et al, 2021). Briefly, *E. coli BL21* cells transformed by the expression vector were cultured in Luria Bertani medium at 37°C. When the OD600 of the culture reached 0.6–0.8, 0.2 mM IPTG was added to induce production of recombinant protein. After further growth for 16 h at 16°C, the cells were harvested by centrifugation, resuspended in buffer A (20 mM Tris and 500 mM NaCl, pH 7.5) and lysed by sonication. The cell debris was removed by centrifugation. For GST-tagged sdAb, the supernatant was loaded onto GST affinity chromatography column (17075601, Glutathione Sepharose 4B; GE Healthcare) which was equilibrated with buffer A. After washed three times with buffer A containing 1% Triton X-100, GST-tagged sdAb was incubated with TEV protease at 4°C for overnight and the supernatant was collected and loaded onto Ni²⁺-NTA affinity column (17-5318-06, Ni Sepharose 6 Fast Flow;

GE Healthcare) for TEV protease removal. For 6× His-tagged TEV protease, SQSTM1 or its truncated mutant proteins purification, the supernatant was loaded onto Ni²⁺-NTA affinity column and washed three times with 50 mM imidazole in buffer A. 6× His-tagged proteins were eluted with buffer A containing 300 mM imidazole. For His pull-down assay, the purified sdAb and 6× His-tagged SQSTM1 or its truncations were mix by the molar ratio of 1/3 in buffer A, then Ni²⁺-NTA agarose beads was added for incubating at 4°C for 2 h, followed by washing three times with buffer A. The pulled-down samples were subjected to SDS–PAGE followed by Western blotting.

### Immunofluorescence and fluorescent microscopy

Flag-tagged *SQSTM1* or its mutants was transiently transduced into HEK293T cells by PEI. After 24 h, the purified sdAb was added into the culture medium. After overnight incubation, the cells were fixed with 1% PFA followed by immunofluorescence staining as previously described (Im et al, 2019). Cells were seeded on cover slip coated with poly-lysine in 24-well plates. After fixed with 1% PFA for 10 min, the cells were washed twice in PBS buffer and PLA or IF was performed. Mouse anti-Flag and rabbit anti-HA or anti-SQSTM1 antibodies were used as primary antibodies for PLA. The purified anti-SQSTM1 sdAb and anti-Flag antibody were used for immunocytochemistry staining. The coverslips were mounted on glass slides using Vecta shield with DAPI and examined using a Leica SP8 laser scanning confocal microscope.

### Co-IP and Western blot

Whole-cell lysates were generated with a hand-held tip homogenizer in RIPA buffer containing complete protease inhibitors cocktail tablet. The lysates were centrifuged at 13,800*g* for 15 min to remove insoluble fractions and the protein concentration was measured with BCA protein assay kit. The supernatants were precleared by protein G-Sepharose CL-4B beads incubated at 4°C for 30 min and centrifuged at 13,800*g* for 10 min. The supernatants were incubated with appropriate primary antibodies or normal mouse IgG at 4°C overnight, followed by addition of protein G-Sepharose CL-4B beads for 2 h at 4°C. The antibody–protein complexes were collected after three washes in RIPA buffer containing protease inhibitors. The samples were resolved by SDS–PAGE and transferred to nitrocellulose membrane. Immunoblot analysis was performed by Western blotting with corresponding antibodies and visualized on Kodak X-ray film using the ECL detection reagents. The purified anti-SQSTM1 sdAb and anti-Flag antibody were also used for conventional Western blot.

---

in *Sqstm1⁻ᐟ⁻* LLC1 cells, respectively. Arrow, h.c., IgG heavy chain or l.c., IgG light chain, respectively. Experiments were performed in triplicate, yielding similar results. Scale bar, 10 μm. **(F)** Immunoblot of LC3B and SQSTM1 in A549 cells treated with anti-SQSTM1 sdAb clone #1 at the indicated concentration for 12 h followed by 50 mM trehalose (Tre) treatment for another 12 h, compared with sdAb Con treatment. ACTB as a loading control. **(G, H)** Immunoblot (G) and IF staining (H) of Sqstm1 using anti-SQSTM1 sdAb clone #2 in *Sqstm1⁻ᐟ⁻* LLC1 cells, compared with sdAb Con. ACTB as a loading control. **(I)** Measurement of the interaction of anti-SQSTM1 sdAb #1 and GST-SQSTM1, 221–440 aa ($K_d$ = 8.80 ± 1.39 μM) by ITC, compared to the negative control of sdAb Con or GST. The upper panels show the heat effect upon titration of 400 μM anti-SQSTM1 sdAb #1 with 40 μM GST-SQSTM1, 221–440 aa. The lower panels represent the binding isotherm and the best–fit curve. Source data are available for this figure.

## ITC

ITC experiments were carried out at 25°C in 1× PBS on a MicroCal PEAQ-ITC (Malvern Microcal). Protein samples were solubilized in 1× PBS storage buffer. About 36.4 µl injections of 400 µM anti-SQSTM1 sdAb clone #1 or sdAb Con (deleted CDR3 as a negative control) were automatically injected into 300 µl of 40 µM GST-SQSTM1, 221–440 aa. Data were integrated and fitted to a single site binding equation using MicroCal PEAQ-ITC Analysis Software which was included with the ITC module (Malvern Microcal).

# Supplementary Information

# Acknowledgements

We thank Drs K Zhang, D Hu, and X Yan (Tianjin Medical University, Tianjin, China) for their useful discussion. This work was funded by grants from the Ministry of Science and Technology of China (Grant No. 2018YFC1313002), the National Natural Science Foundation of China (81825017, 81773034, 81872350, and 8217113342) and the Tianjin Municipal Science and Technology Commission (18JCZDJC99100).

## Author Contributions

Y Yin: data curation and formal analysis.
F Yan: data curation and formal analysis.
R Zhou: data curation and formal analysis.
M Li: data curation.
J Ma: data curation.
Z Liu: conceptualization, data curation, supervision, investigation, and writing—original draft.
Z Ma: conceptualization, data curation, supervision, investigation, methodology, project administration, and writing—review and editing.

## Conflict of Interest Statement

The authors declare that they have no conflict of interest.

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
