## [Reviewer comments · Life Science Alliance]

Life Science Alliance

Single-domain antibody screening by isPLA-seq

Yueyuan Yin, Fei Yan, Ruimin Zhou, Mingchen Li, Jinyi Ma, Zhe Liu, and Zhenyi Ma

DOI: <https://doi.org/10.26508/lsa.202101115>

Corresponding author(s): Zhenyi Ma, Tianjin Medical University and Zhe Liu, Tianjin Medical University

Review Timeline:

Submission Date:	2021-05-07
Editorial Decision:	2021-06-17
Revision Received:	2021-09-03
Editorial Decision:	2021-10-01
Revision Received:	2021-10-11
Editorial Decision:	2021-10-12
Revision Received:	2021-10-13
Accepted:	2021-10-14

Transaction Report:

June 17, 2021

Re: Life Science Alliance manuscript #LSA-2021-01115-T

Dr. Zhenyi Ma
Tianjin Medical University
Biochemistry and Molecular Biology
22 Qixiangtai Road
Heping District
Tianjin, Tianjin 300070
CHINA

Dear Dr. Ma,

Thank you for submitting your manuscript entitled "Single-domain antibody screening by isPLA-seq" to Life Science Alliance. The manuscript was assessed by an expert reviewer, whose comments are appended to this letter. We invite you to submit a revised manuscript addressing the Reviewer comments.

When submitting the revision, please include a letter addressing the reviewer comments point by point.

Thank you for this interesting contribution to Life Science Alliance. We are looking forward to receiving your revised manuscript.

Sincerely,

-- A letter addressing the reviewer comments point by point.

B. MANUSCRIPT ORGANIZATION AND FORMATTING:

Reviewer #1 (Comments to the Authors (Required)):

Review report:

The manuscript "Single-domain antibody screening by isPLA-seq" by Yin et al. provides a new approach for screening a single-domain antibody library where the CDR3 domain is varied, using the in situ proximity ligation assay (isPLA) for selecting cells transfected with constructs encoding binding reagents via fluorescence-activated cell sorting (FACS).

The isPLA method is commonly used for localized detection of proteins, protein-protein interactions, and post-translational modifications, using the rolling cycle amplification for localized signal amplification. The method provided in this manuscript used isPLA to screen sdAbs expressed from the CDR3 cDNA expression library that targets the SQSTM1 protein, which is expressed from another cDNA construct co-transfected with the sdAb library. The authors go on the subject of the two selected clones to functional validation.

The method provided here can be potentially useful as a means to screen for binding reagents without a need to first produce and purify the target protein since the isPLA mechanism allows identification of reagents that bind in proximity to the expressed target protein.

Below are my comments for this manuscript:

Major comments:

1. Is the CABBCII10-CDR3 null construct (replaced with a spacer) a good control construct? Why not using the original CABBCII10 CDR3 sequence as control which has the same length as the degenerated CDR3 region (The CABBCII10 nanobody shouldn't bind to the SQSTM1)? The CABBCII10-CDR3 null construct might have very distinct structure due to lacking the CDR3 domain.
2. The selection of the clones for further characterization is based on the frequency with which sequences are recovered from the isPLA positive cells isolated by FACS. The isPLA signals represent interactions of the sdAb with the SQSTM1. No data supporting why the plasmid sequence recovery frequency is a good criterion for selecting the clones. Other factors might affect the recovery rate, like the transfection efficiency, PCR, subcloning, et al.. Can the recovery rate reflect the quality (affinity and selectivity) of the clone? This should be a critical question to answer to make this method valid, otherwise, the selection of clone 1 and 2 would be quite arbitrary.
3. The affinity is a critical feature for binders, it is a pity that there is no data on the affinities and the affinity comparison of selected clones. If the affinities can be measured and compared between several clones, it would be a piece of valuable information for checking the correlation between the quality of the clones with sequence recovery frequencies.
4. It would be beneficial if the authors can add more background information for other screening methods to the introduction section and comparing the advantages or limitations of the method compared with other available screening methods in the discussion section, so the readers can know the advantages of the presented method compared to others.
5. Compared with figure 3E, H, the isPLA signals in figure 1C are too tiny. The normal isPLA signal should be around 1 micrometer in diameter with the protocol provided by the authors. Since the SQSTM1 protein expresses in the cytosol, the isPLA signals should also be located in the cytosol rather than in nuclei. I suspect the signal in this image is the background of the isPLA staining rather than the true isPLA signal. The true signals should like the signals in Figure 3E,H.
6. The overall writing of the manuscript needs to be checked carefully and improved.

Minor comments:

1. "...provide a broad spectrum to identify..." (page 1) is not clear to me.
2. "high-sensitive throughput" may change to "high-sensitive, high-throughput"
3. Page 2 line 5, ...an sdAb library.... Should be .. a sdAb library...
4. As shown in Figure 3 F and 3G, several western images seem composed of several strips of images. The authors may need to explain and provide better images. (I put this item here in the minor comments section because the images could be distorted by file format transformation)
5. "sdAb clone #1 showed significant effects on autophagic flux", there is no statistical analysis data to show the change of the flux, it is not proper to use "significant".
6. The name of the method isPLA-seq appeared only in the title, abstract, and one of the figure legends. The method definition was not explained in the text at all.

9-3-2021

Re: Life Science Alliance manuscript #LSA-2021-01115-TR

Dear Dr. Sawey and the LSA editorial group,

We are very pleased to resubmit our manuscript “Single-domain antibody screening by *isPLA-seq*” to Life Science Alliance. Following the suggestions of the reviewer, we have performed additional experimental data to clarify some results with appropriate explanations. We also made some modifications and improvements to our manuscript. All of these changes are highlighted in red color in the revised manuscript and a detailed point-by-point response is appended below.

Responses to Reviewer Comments

Reviewer #1 (Comments to the Authors (Required)):

Review report:

The manuscript "Single-domain antibody screening by *isPLA-seq*" by Yin et al. provides a new approach for screening a single-domain antibody library where the CDR3 domain is varied, using the in situ proximity ligation assay (*isPLA*) for selecting cells transfected with constructs encoding binding reagents via fluorescence-activated cell sorting (FACS).

The *isPLA* method is commonly used for localized detection of proteins, protein-protein interactions, and post-translational modifications, using the rolling cycle amplification for localized signal amplification. The method provided in this manuscript used *isPLA* to screen sdAbs expressed from the CDR3 cDNA expression library that targets the SQSTM1 protein, which is expressed from another cDNA construct co-transfected with the sdAb library. The authors go on the subject of the two selected clones to functional validation.

The method provided here can be potentially useful as a means to screen for binding reagents without a need to first produce and purify the target protein since the *isPLA* mechanism allows identification of reagents that bind in proximity to the expressed target protein.

Response: We are grateful to the reviewer for the detailed comments and suggestions for improvement. We believe that the *isPLA-seq* described in this manuscript is a high-sensitive and high-throughput screening tool for protein-protein interaction including the antigen-antibody binding. It may also extend our previous work in sdAb, protein interactome and, more broadly, any biomolecule field in cells.

Below are my comments for this manuscript:

Major comments:

1. Is the CABBCII10-CDR3 null construct (replaced with a spacer) a good control

construct? Why not using the original CABBCII10 CDR3 sequence as control which has the same length as the degenerated CDR3 region (The CABBCII10 nanobody shouldn't bind to the SQSTM1)? The CABBCII10-CDR3 null construct might have very distinct structure due to lacking the CDR3 domain.

Response: This is a really helpful suggestion and we thank the reviewer for pointing this out. Camel single-domain antibody framework of CABBCII10 has been shown as a potential candidate for the exchange of antigen specificity by CDR3 grafting (J Mol Biol. 2005;352(3):597-607. Chemistry 2006;12:1915-1923.). CDR3 is the main contributor for epitope recognition, while CDR1 and CDR2 regions stabilize the sdAb structure and assist in the binding strength as previously demonstrated (J Mol Biol. 2018;430(21):4369-86. Proteins. 2018;86(7):697-706.).

In response to the reviewer's comment, we used CABBCII10 CDR3 sequence (VRGYFMRLPSSHNFY) as another sdAb control in our experimental work. As expected, CABBCII10 as well the sdAb Con (CABBCII10-CDR3 null) is negative to recognize SQSTM1 full length as shown in Fig. 3 B by *isPLA*, suggesting CABBCII10 does not bind to SQSTM1. These sdAb Cons might still have some distinct structure due to lacking the CDR3 domain or its replacement by the spacer we used in this study, however, they loss their specificity of recognizing SQSTM1 *in vitro* and *in vivo* (Figure 3B and C). Although these negative controls did not show the binding activity to SQSTM1, it might still keep their nonspecific binding activity to interact with the intracellular proteins with low affinity.

2. The selection of the clones for further characterization is based on the frequency with which sequences are recovered from the *isPLA* positive cells isolated by FACS. The *isPLA* signals represent interactions of the sdAb with the SQSTM1. No data supporting why the plasmid sequence recovery frequency is a good criterion for selecting the clones. Other factors might affect the recovery rate, like the transfection efficiency, PCR, subcloning, et al.. Can the recovery rate reflect the quality (affinity and selectivity) of the clone? This should be a critical question to answer to make this method valid, otherwise, the selection of clone 1 and 2 would be quite arbitrary.

Response: We agree that this is an important feature of the *isPLA*-seq method. As a screening method, the factors including transfection efficiency, PCR, subcloning, et al. definitely affect the sdAb recovery rate. Technically, we used the enriched subclones for the next round screening of the sdAb against specific epitope. This kind of procedure enhances the enrichment of the specific sdAbs against the target. In our revised manuscript, we mainly focus on the principle of this protocol for sdAb screening and identification. After each round of *isPLA*-seq screening, we may compare the CDR3 sequences showing up in all of the sequences and may figure out the real sdAb candidates for a specific epitope. Lastly, we also validate clone #1 and #2 of their specificity *in vitro* and *in vivo*. In this case, we could not rule out the rest clones may have distinct affinity to SQSTM1

due to their corresponding epitopes.

3. The affinity is a critical feature for binders, it is a pity that there is no data on the affinities and the affinity comparison of selected clones. If the affinities can be measured and compared between several clones, it would be a piece of valuable information for checking the correlation between the quality of the clones with sequence recovery frequencies.

Response: We agree with the reviewer that these are important points. Initially, we found the recombinant SQSTM1 protein formed inclusion body in *E. coli* BL21 cells, thus it is hard to get large amount of recombinant SQSTM1 or its truncated proteins for *in vitro* binding assay such as ITC. Then we used GST-fused SQSTM1, 221-440 aa to test its binding affinity to clone #1 by isothermal titration calorimetry (ITC). We found the dissociation constant (K_d) was $8.80 \pm 1.39 \mu\text{M}$, compared to the non-binding controls (Fig. 3 I). In the revised manuscript we have provided the experimental data showing this affinity feature for the recombinant sdAb clone #1 and SQSTM1 (221-440 aa).

4. It would be beneficial if the authors can add more background information for other screening methods to the introduction section and comparing the advantages or limitations of the method compared with other available screening methods in the discussion session, so the readers can know the advantages of the presented method compared to others.

Response: This is a great suggestion. We have included more background information for other screening methods such as yeast-two hybridization and phage display under Introduction section (Page 2, lines 37-41) and comparing the advantages or limitations of the method compared with other available screening methods under Discussion session on Page 5, lines 119-124 as well. The added information greatly expanded the understanding the advantages and disadvantages of the screening approaches for specificity of antibodies.

5. Compared with figure 3E, H, the isPLA signals in figure 1C are too tiny. The normal isPLA signal should be around 1 micrometer in diameter with the protocol provided by the authors. Since the SQSTM1 protein expresses in the cytosol, the isPLA signals should also be located in the cytosol rather than in nuclei. I suspect the signal in this image is the background of the isPLA staining rather than the true isPLA signal. The true signals should like the signals in Figure 3E,H.

Response: We thank the reviewer for pointing this out. These images were taken under confocal after the first fluorescence-activated cell sorting (FACS) of the positive cells which may show the weak dots fluorescence compared with the isPLA positive signals in Figure 3E or H. We have replaced these images with better resolution and showed that the isPLA positive signals of SQSTM1 and its sdAb candidates were predominantly cytoplasmic as

well as nuclear.

Since SQSTM1/p62 contains one nuclear export signal (NES, 303-320 aa,) and two nuclear localization signals (NLS, 186-189 aa and 264-267 aa) as shown in Figure 3A, variable nuclear and cytosolic localization of SQSTM1 has been noticed and SQSTM1 shuttles between the nucleus and the cytoplasm (J Biol Chem. 2010;285(8):5941-53. Vet Pathol. 2015;52(4):621-30.). In the nucleus, p62 is thought to recruit proteasomes to nuclear polyubiquitinated protein aggregates and can even export ubiquitinated substrates from the nucleus into the cytosol, where autophagy offers a more robust degradative capacity (Anticancer Res. 2019;39(12):6711-6722. Nat Commun. 2020;11(1):2306.). However, additional studies are needed to further characterize SQSTM1 functionality of its nuclear and cytoplasmic localization precisely.

6. The overall writing of the manuscript needs to be checked carefully and improved.

Response: We have performed the language editing by a native English speaker and have proofread our revised manuscript carefully including the grammatical errors.

Minor comments:

1. "...provide a broad spectrum to identify..." (page 1) is not clear to me.

Response: Thanks the reviewer for pointing this out. We have now clarified this sentence as follows, "This method provides a general means to identify the functional measure of sdAb and its complementary epitopes and its potential applications to investigate cellular processes", on Page 2, lines 26-28.

2. "high-sensitive throughput" may change to "high-sensitive, high-throughput"

Response: It has been done in the revised version of this manuscript as suggested. We thank the reviewer for this excellent suggestion.

3. Page 2 line 5, ...an sdAb library.... Should be .. a sdAb library...

Response: We thank the reviewer for this suggestion. We have now corrected all of them throughout the whole revised version of this manuscript.

4. As shown in Figure 3 F and 3G, several western images seem composed of several strips of images. The authors may need to explain and provide better images. (I put this item here in the minor comments section because the images could be distorted by file format transformation)

Response: Thanks for this suggestion. We have rescanned the original Western blot images in Figure 3F and 3G, and replaced the images in these two panels. The original scanned images of these data have also been uploaded as supplementary "Source Data".

5. "sdAb clone #1 showed significant effects on autophagic flux", there is no statistical analysis data to show the change of the flux, it is not proper to use "significant".

Response: We agree with the reviewer and this is an important issue based on our data without statistical analysis. We have corrected this sentence as follows on Page 4, lines 97-98, "...we tested whether sdAb clone #1 against SQSTM1 disturbs autophagic flux".

6. The name of the method isPLA-seq appeared only in the title, abstract, and one of the figure legends. The method definition was not explained in the text at all.

Response: The reviewer's comment is well taken. Now we have explained the definition of *isPLA-seq* as follows, "Using a high-throughput *in situ* proximity ligation assay (*isPLA*)-based cDNA screening of sdAb-epitope interactions in single cells followed by sequencing (namely *isPLA-seq*), here we outlined a sdAb screening approach for SQSTM1 (Fig. 1 A)", on Page 3, lines 52-54.

In sum, we would like to thank the reviewer for his/her thoughtful comments and advice on revision of this manuscript. After addressing these comments, we believe that these modifications have improved the *isPLA-seq* methods, which might be broadly used in future single-domain antibody research or its related fields.

Sincerely yours,

Zhenyi Ma

Zhe Liu

Tianjin Medical University

Email: zhyma@tmu.edu.cn or zheliu@tmu.edu.cn

October 1, 2021

Re: Life Science Alliance manuscript #LSA-2021-01115-TR

Dr. Zhenyi Ma
Tianjin Medical University
Biochemistry and Molecular Biology
22 Qixiangtai Road
Heping District
Tianjin, Tianjin 300070
China

Dear Dr. Ma,

Thank you for submitting your revised manuscript entitled "Single-domain antibody screening by isPLA-seq" to Life Science Alliance. The manuscript has been seen by the original reviewers whose comments are appended below. While the reviewers continue to be overall positive about the work in terms of its suitability for Life Science Alliance, a key question about whether or not the recovery rate reflects the affinity was not fully addressed.

Our general policy is that papers are considered through only one revision cycle; however, given that the suggested changes are relatively minor, we are open to one additional short round of revision. Please note that I will expect to make a final decision without additional reviewer input upon resubmission.

Please submit the final revision within one month, along with a letter that includes a point by point response to the remaining reviewer comments.

To upload the revised version of your manuscript, please log in to your account: <https://lsa.msubmit.net/cgi-bin/main.plex>
You will be guided to complete the submission of your revised manuscript and to fill in all necessary information.

B. MANUSCRIPT ORGANIZATION AND FORMATTING:

Sincerely,

Reviewer #1 (Comments to the Authors (Required)):

Thanks to the authors for their efforts to respond to the comments. And I feel it is a very unique application for isPLA with the potential for screening protein binders. The validity of the method is based on the assumption that there is a correlation between the affinity of the clones and the recovery rate of the sequences.

But my concern is still there:

Can the recovery rate reflect the affinity?

I aware the authors have tested the clone 1 and 2 for functional validation. but I still feel the selection of clones 1 and 2 is quite arbitrary without showing the correlation of recovery rate and affinity. So potential users might be facing the same problem.

It is very possible that some non-binding constructs still got amplified by entering the same cells with constructs can bind to target, so all of them will be amplified during PCR, re-cloned, and transfected to cells. so I wonder whether it is enough to take away all non-bound constructs by three round of selection (this is not to suggest the authors to do more round of selection). If there is a correlation between recovery rate with affinity (higher affinity correlate with higher recovery rate?), it will be easy for selecting the best constructs after several rounds of selection by setting a cutoff.

And among 27 sequences that appeared in the 260 sequences, 10 clones have frequency over 10/260. Plus the clone 1 used for Kd characterization has a quite low affinity (micromolar range), if there is no other criterion for selecting the best sequence among the 27, it will be still a tremendous amount of work to find out the best clones to work with, among the 27 sequences, which will limit the advantages of this method.

Other comments:

1. The focus of the paper is to introduce a new method for screening the sdAbs, but a large part of the discussion is focusing on the advantages of sdAbs.

2. One question out of curiosity, I am not working on sdAbs, but one question is that could CDR3 have enough affinity for targets? Based on the paper below, it seems most sdARs have at least two CDRs involved in binding.
(<https://www.frontiersin.org/articles/10.3389/fimmu.2017.00977/full>)

10-11-2021

Re: Life Science Alliance manuscript #LSA-2021-01115-TRR

Dear Dr. Sawey and the LSA editorial group,

We are very pleased to resubmit our manuscript “Single-domain antibody screening by *isPLA-seq*” to Life Science Alliance.

We understand the Reviewer’s key question about whether or not the recovery rate reflects the affinity, which is not easily and fully addressed *via* experimental work. The critical point of the *isPLA-seq* method is the interactions by *isPLA* are derived from the proximal contact in single cells. This may reflect the high sensitivity of our screening method. However, all of the *in vitro* assays including pulldown and ITC show the direct interaction of two components only, which is unnecessarily identical to the events in living cells. Because of the dynamic interfaces of a given protein complexes in cells, we have no conclusive statement on the interactions which are the same tendency to integrate the *in vitro* and the *in vivo* data. However, the collected information indeed provides us useful clues for the sdAb’s functionality both *in vitro* and *in vivo*, as we addressed in this revised manuscript.

Following the suggestions of the editor and the reviewer, we have made some modifications and improvements to our manuscript. All of these changes are highlighted in red color in the revised manuscript and a detailed point-by-point response is appended below.

Responses to Reviewer Comments

Reviewer #1 (Comments to the Authors (Required)):

Thanks to the authors for their efforts to respond to the comments. And I feel it is a very unique application for *isPLA* with the potential for screening protein binders. The validity of the method is based on the assumption that there is a correlation between the affinity of the clones and the recovery rate of the sequences.

Response: We totally agree with this reviewer and we still believe that the *isPLA-seq* described here is a high-sensitive and high-throughput screening method for protein-protein interactions (PPIs).

But my concern is still there:

Comment 1. Can the recovery rate reflect the affinity?

Response: Actually, the answer is “maybe” or “maybe not”. It depends on the conformation of the epitopes on target protein and the potential candidates in the sdAb library. In our on-going screening work of the different bait proteins, we found one clone continuously showed up with relatively higher recovery rate (~20%) in some screens, and was absent in the others. However, it still specifically recognizes its bait protein in pulldown assay

in vitro (unpublished data). It is possible that the similar conformation on the different bait proteins may still keep their epitope property during the *isPLA* screening in single cells, resulting in a “common” clone. Similarly, it happens upon using a “fussy” antibody, we usually get some non-specific protein bands in Western blot assay.

Comment 2. I aware the authors have tested the clone 1 and 2 for functional validation. but I still feel the selection of clones 1 and 2 is quite arbitrary without showing the correlation of recovery rate and affinity. So potential users might be facing the same problem.

Response: We totally agree with this reviewer. After the high-through sequencing of the recovered clones, we analyzed the predicted docking data of all the recovered CDR3 regions and their recognized epitopes using HDOCK (Nat Protoc. 2020 May;15(5):1829-1852.). Based on these predictive data, we chose clone #1 and #2 sdAbs with higher docking score for the following experimental validation. Indeed, these docking data provided us useful information such as the docking regions on sdAb as well as on SQSTM1. However, these data provides the potential rather than the exact precision, as the web server states. We have now added this information as follows, “After high-through sequencing, docking of the recovered sdAbs and SQSTM1 molecules was conducted using HDOCK as described previously (Yan et al, 2020).”, on Page 10, lines 241-3.

Comment 3. It is very possible that some non-binding constructs still got amplified by entering the same cells with constructs can bind to target, so all of them will be amplified during PCR, re-cloned, and transfected to cells. so I wonder whether it is enough to take away all non-bound constructs by three round of selection (this is not to suggest the authors to do more round of selection). If there is a correlation between recovery rate with affinity (higher affinity correlate with higher recovery rate?), it will be easy for selecting the best constructs after several rounds of selection by setting a cutoff.

Response: Again, this is related to the key comment by this reviewer. We agree that the non-specific binding candidates may not be excluded even after more rounds of screenings, because of the sensitivity of *isPLA*-seq method we described here. However, we found the HDOCK or the other predictive molecular docking software might help figuring out this problematic issue. Indeed, using these docking web servers, we may predict the structures based on the interaction predictions between the recovered sdAbs and their epitopes on SQSTM1. Based on these predictions, we focused on the potential candidates for their following validation. In this case, higher affinity may not correlate with higher recovery rate, as explained in Response to Comment 1.

Comment 4. And among 27 sequences that appeared in the 260 sequences, 10 clones

have frequency over 10/260. Plus the clone 1 used for Kd characterization has a quite low affinity (micromolar range), if there is no other criterion for selecting the best sequence among the 27, it will be still a tremendous amount of work to find out the best clones to work with, among the 27 sequences, which will limit the advantages of this method.

Response: We agree that this is an important feature of the *is*PLA-seq method followed by *in vitro* validation. We know that the recombinant SQSTM1 protein formed inclusion body in *E. coli* BL21 cells, thus it is hard to get large amount of recombinant full-length SQSTM1 or its truncated proteins for *in vitro* isothermal titration calorimetry (ITC) assay, as we explained previously. However, anti-SQSTM1 sdAb clone #1 indeed disturbs autophagic flux in A549 cells (Fig. 3 F), which are SQSTM1 (full length)-positive cells. We also realized that the other analysis approaches such as the molecular docking prediction may provide some helpful information, which is what we did before (see Response to Comment 2). Based on the predicted structures, we chose those candidate clones for their validation using experimental tools such as *is*PLA, Co-IP, and pulldown. Using these techniques in our lab, we spent less than three weeks to validate these two sdAbs in this manuscript.

Other comments:

1. The focus of the paper is to introduce a new method for screening the sdAbs, but a large part of the discussion is focusing on the advantages of sdAbs.

Response: We restate the advantages of our sdAb screen method as follows on Pages 5-6, line 124-9, “The high sensitive, high-throughput sdAb library screening toolkit for *in situ* antibody-epitope recognitions described here provides several advantages as follows: (a) sdAbs prediction directly from the recovered CDR3 DNA sequences; (b) visualization and individual quantification of subcellular distribution of sdAb-epitope recognitions; (c) easy validation and reevaluation of sdAb functionality, and (d) potential recognition both native and denature epitopes.”

2. One question out of curiosity, I am not working on sdAbs, but one question is that could CDR3 have enough affinity for targets? Based on the paper below, it seems most sdARs have at least two CDRs involved in binding. (<https://www.frontiersin.org/articles/10.3389/fimmu.2017.00977/full>)

Response: We agree with this reviewer. More than 4 CDRs in some sdAbs may be needed depending on the epitope recognizing region on the target proteins (Front Immunol. 2017 Aug 21;8:977.). However, the smaller number of CDRs does not seem to lower the affinity of sdAbs to bind their epitopes. As shown, the framework of our sdAb library is derived from CABBCII10, which has been shown as a potential candidate for the exchange of antigen specificity by CDR3 grafting (J Mol Biol. 2005;352(3):597-607. Chemistry 2006;12:1915-1923.). And the CDR1

and CDR2 regions stabilize the sdAb structure and assist in the binding strength as previously demonstrated (J Mol Biol. 2018;430(21):4369-86. Proteins. 2018;86(7):697-706.). In this case, we screened and validated 2 sdAbs against SQSTM1 with different usages, functional interference in cells and *in vitro* detection, such as Western blotting and immunofluorescence (IF) assays.

In sum, we would like to thank the editor for his supportive comments and advice on the revision of this manuscript. After addressing these comments, we believe that these modifications have improved the *is*PLA-seq method, which might be broadly used in future sdAb research and its related fields.

Sincerely yours,

Zhenyi Ma
Zhe Liu
Tianjin Medical University
Email: zhyma@tmu.edu.cn or zheliu@tmu.edu.cn

October 12, 2021

RE: Life Science Alliance Manuscript #LSA-2021-01115-TRR

Dr. Zhenyi Ma
Tianjin Medical University
Biochemistry and Molecular Biology
22 Qixiangtai Road
Heping District
Tianjin, Tianjin 300070
China

Dear Dr. Ma,

Thank you for submitting your revised manuscript entitled "Single-domain antibody screening by isPLA-seq". We would be happy to publish your paper in Life Science Alliance pending final revisions necessary to meet our formatting guidelines.

- please add ORCID ID for secondary corresponding author-they should have received instructions on how to do so
- please add the Twitter handle of your host institute/organization as well as your own or/and one of the authors in our system
- please use the [10 author names, et al.] format in your references (i.e. limit the author names to the first 10)
- tables must be in editable .doc or excel format
- please use capital letters when labeling panels in actual figures

A. FINAL FILES:

B. MANUSCRIPT ORGANIZATION AND FORMATTING:

**Submission of a paper that does not conform to Life Science Alliance guidelines will delay the acceptance of your

manuscript.**

The license to publish form must be signed before your manuscript can be sent to production. A link to the electronic license to publish form will be sent to the corresponding author only. Please take a moment to check your funder requirements.

Sincerely,

October 14, 2021

RE: Life Science Alliance Manuscript #LSA-2021-01115-TRRR

Dr. Zhenyi Ma
Tianjin Medical University
Biochemistry and Molecular Biology
22 Qixiangtai Road
Heping District
Tianjin, Tianjin 300070
China

Dear Dr. Ma,

Thank you for submitting your Research Article entitled "Single-domain antibody screening by isPLA-seq". It is a pleasure to let you know that your manuscript is now accepted for publication in Life Science Alliance. Congratulations on this interesting work.

DISTRIBUTION OF MATERIALS:

Again, congratulations on a very nice paper. I hope you found the review process to be constructive and are pleased with how the manuscript was handled editorially. We look forward to future exciting submissions from your lab.

Sincerely,
